# Explaining Deep Learning Models – A Bayesian Non-parametric Approach

**Wenbo Guo**
The Pennsylvania State University
wzg13@ist.psu.edu

**Sui Huang**
Netflix Inc.
shuang@netflix.com

**Yunzhe Tao**
Columbia University
y.tao@columbia.edu

**Xinyu Xing**
The Pennsylvania State University
xxing@ist.psu.edu

**Lin Lin**
The Pennsylvania State University
llin@psu.edu

## Abstract

Understanding and interpreting how machine learning (ML) models make decisions have been a big challenge. While recent research has proposed various technical approaches to provide some clues as to how an ML model makes individual predictions, they cannot provide users with an ability to inspect a model as a complete entity. In this work, we propose a novel technical approach that augments a Bayesian non-parametric regression mixture model with multiple elastic nets. Using the enhanced mixture model, we can extract generalizable insights for a target model through a global approximation. To demonstrate the utility of our approach, we evaluate it on different ML models in the context of image recognition. The empirical results indicate that our proposed approach not only outperforms the state-of-the-art techniques in explaining individual decisions but also provides users with an ability to discover the vulnerabilities of the target ML models.

## 1 Introduction

When comparing with relatively simple learning techniques such as decision trees and K-nearest neighbors, it is well acknowledged that complex learning models – particularly, deep neural networks (DNN) – usually demonstrate superior performance in classification and prediction. However, they are almost completely opaque, even to the engineers that build them [20]. Presumably as such, they have not yet been widely adopted in critical problem domains, such as diagnosing deadly diseases [13] and making million-dollar trading decisions [14].

To address this problem, prior research proposes to derive an interpretable explanation for the output of a DNN. With that, people could understand, trust and effectively manage a deep learning model. From a technical prospective, this can be interpreted as pinpointing the most important features in the input of a deep learning model. In the past, the techniques designed and developed primarily focus on two kinds of methods – (1) *whitebox explanation* that derives interpretation for a deep learning model through forward or backward propagation approach [26, 36], and (2) *blackbox explanation* that infers explanations for individual decisions through local approximation [21, 23]. While both demonstrate a great potential to help users interpret an individual decision, they lack an ability to extract insights from the target ML model that could be generalized to future cases. In other words, existing methods could not shed lights on the general sensitivity level of a target model to specific input dimensions and hence fall short in foreseeing when prediction errors might occur for future cases.

In this work, we propose a new technical approach that not only explains an individual decision but, more importantly, extracts generalizable insights from the target model. As we will show in

Section 4, we define such insights as the *general sensitivity level of a target model to specific input dimensions*. We demonstrate that model developers could use them to identify model strengths as well as model vulnerabilities. Technically, our approach introduces multiple elastic nets to a Bayesian non-parametric regression mixture model. Then, it utilizes this model to approximate a target model and thus derives its generalizable insight and explanation for its individual decision. The rationale behind this approach is as follows.

A Bayesian non-parametric regression mixture model can approximate arbitrary probability density with high accuracy [22]. As we will discuss in Section 3, with multiple elastic nets, we can augment a regression mixture model with an ability to extract patterns (generalizable insights) even from a learning model that takes as input data of different extent of correlations. Given the pattern, we could extrapolate input features that are critical to the overall performance of an ML model. This information can be used to facilitate one to scrutinize a model's overall strengths and weaknesses. Besides extracting generalizable insights, the proposed model can also provide users with more understandable and accountable explanations. We will demonstrate this characteristic in Section 4.

## 2 Related Work

Most of the works related to model interpretation lie in demystifying complicated ML models through *whitebox* and *blackbox* mechanisms. Here, we summarize these works and discuss their limitations. It should be noticed that we do not include those works that identify training samples that are most responsible for a given prediction (*e.g.*, [12, 15]) and those works that build a self-interpretable deep learning model [7, 33].

The whitebox mechanism augments a learning model with the ability to yield explanations for individual predictions. Generally, the techniques in this kind of mechanism follow two lines of approaches – ❶ occluding a fraction of a single input sample and identifying what portions of the features are important for classification [4, 6, 17, 36, 37], and ❷ computing the gradient of an output with respect to a given input sample and pinpointing what features are sensitive to the prediction of that sample [1, 8, 24, 25, 26, 29, 32]. While both can give users an explanation for a single decision that a learning model reach, they are not sufficient to provide a global understanding of a learning model, nor capable of exposing its strengths and weaknesses. In addition, they typically cannot be generally applied to explaining prediction outcomes of other ML models because most of the techniques following this mechanism are designed for a specific ML model and require altering that learning model.

The blackbox mechanism treats an ML model as a black box, and produces explanations by locally learning an interpretable model around a prediction. For example, LIME [23] and SHAP [21] are the same kind of explanation techniques that sample perturbed instances around a single data sample and fit a linear model to perform local explanations. Going beyond the explanation of a single prediction, they both can be extended to explain the model as a complete entity by selecting a small number of representative individual predictions and their explanations. However, explanations obtained through such approaches cannot describe the full mapping learned by an ML model. In this work, our proposed technique derives a generalizable insight directly from a target model, which provides us with the ability to unveil model weaknesses and strengths.

## 3 Technical Approach

### 3.1 Background

A Bayesian non-parametric regression mixture model (*i.e.*, mixture model for short) consists of multiple Gaussian distributions:

$$y_i|\mathbf{x}_i, \mathbf{\Theta} \sim \sum_{j=1}^{\infty} \pi_j N(y_i \mid \mathbf{x}_i \boldsymbol{\beta}_j, \sigma_j^2), \tag{1}$$

where $\mathbf{\Theta}$ denotes the parameter set, $\mathbf{x}_i \in \mathbb{R}^p$ is the $i$-th data sample of the sample feature matrix $\mathbf{X}^{\mathrm{T}} \in \mathbb{R}^{p \times n}$, and $y_i$ is the corresponding prediction in $\mathbf{y} \in \mathbb{R}^n$, which is the predictions of $n$ samples. $\pi_{1:\infty}$ are the probabilities tied to the distributions with the sum equal to 1, and $\boldsymbol{\beta}_{1:\infty}$ and $\sigma_{1:\infty}^2$ represent the parameters of regression models, with $\boldsymbol{\beta}_j \in \mathbb{R}^p$ and $\sigma_j^2 \in \mathbb{R}$.

In general, model (1) can be viewed as a combination of infinite number of regression models and be used to approximate any learning model with high accuracy. Given a learning model $g : \mathbb{R}^p \to \mathbb{R}$, we can therefore approximate $g(\cdot)$ with a mixture model using $\{\mathbf{X}, \mathbf{y}\}$, a set of data samples as well as their corresponding predictions obtained from model $g$, i.e., $y_i = g(\mathbf{x}_i)$. For any data sample $\mathbf{x}_i$, we can then identify a regression model $\hat{y}_i = \mathbf{x}_i \boldsymbol{\beta}_j + \epsilon_i$, which best approximates the local decision boundary near $\mathbf{x}_i$[1].

Note that in this paper, we assume that a single mixture component is sufficient to approximate the local decision boundary around $\mathbf{x}_i$. Despite the assumption doesnot hold in some cases, the proposed model can be relaxed and extended to deal with these cases. More specifically, instead of directly assigning each instance to one mixture component, we can assign an instance at a mode level [10], (i.e., assigning the instance to a combination of multiple mixture components). When explaining a single instance, we can linearly combine the corresponding regression coefficients in a mode.

Recent research [23] has demonstrated that such a linear regression model can be used for assessing how the feature space affects a decision by inspecting the weights (model coefficients) of the features present in the input. As a result, similar to prior research [23], we can take this linear regression model to pinpoint the important features and take them as an explanation for the corresponding individual decision.

In addition to model approximation and explanation mentioned above, another characteristic of a mixture model is that it can enable multiple training samples to share the same regression model and thus preserve only dominant patterns in data. With this, we can significantly reduce the amount of explanations derived from training data and utilize them as the generalizable insight of a target model.

## 3.2 Challenge and Technical Overview

Despite the great characteristics of a mixture model, it is still challenging for us to use it for deriving generalizable insights or individual explanation. This is because a regression mixture model does not always guarantee a success in model approximation, especially when it deals with samples with diverse feature correlations and data sparsity.

To tackle this challenge, an instinctive reaction is to introduce an elastic net to a Bayesian regression mixture model. Past research [9, 18, 38] has demonstrated that an elastic net encourages the grouping effects among variables so that highly correlated variables tend to be in or out of a mixture model together. Therefore, it can potentially augment the aforementioned method with the ability of dealing with the situation where the features of a high dimensional sample are highly correlated. However, a key limitation of this approach could manifest, especially when it deals with samples with diverse feature correlation and data sparsity.

In the following, we address this issue by establishing a dirichlet process mixture model with multiple elastic nets (DMM-MEN). Different from previous research [35], our approach allows the regularization terms to has the flexibility to reduce a lasso or ridge under some sample categories, while maintaining the properties of the elastic net under other categories. With the multiple elastic nets, the model is able to capture the different levels of feature correlation and sparsity in the data. the In the following, we provide more details of this hierarchical Bayesian non-parametric model.

## 3.3 Technical Details

**Dirichlet Process Regression Mixture Model.** As is specified in Equation (1), the amount of Gaussian distributions is infinite, which indicates that there are infinite number of parameters that need to be estimated. In practice, however, the amount of available data samples is limited and therefore it is necessary to restrict the number of distributions. To do this, truncated Dirichlet process prior [11] can be applied, and Equation (1) can be written as

$$y_i | \mathbf{x}_i, \boldsymbol{\Theta} \sim \sum_{j=1}^{J} \pi_j N(y_i \mid \mathbf{x}_i \boldsymbol{\beta}_j, \sigma_j^2). \tag{2}$$

Where J is the hyper-parameter that specify the upper bound of the number of mixture components. To estimate the parameters $\boldsymbol{\Theta}$, a Bayesian non-parametric approach first models $\pi_{1:J}$ through a "stick-breaking" prior process. With such modeling, parameters $\pi_{1:J}$ can then be computed by

$$\pi_j = u_j \prod_{l=1}^{j-1} (1 - u_l) \quad \text{for } j = 2, ..., J-1, \tag{3}$$

with $\pi_1 = u_1$ and $\pi_J = 1 - \sum_{l=1}^{J-1} \pi_l$. Here, $u_l$ follows a beta prior distribution, Beta$(1, \alpha)$, parameterized by $\alpha$, where $\alpha$ can be drawn from Gamma$(e, f)$ with hyperparameters $e$ and $f$. To make the computation efficient, $\sigma_j^2$ is set to follow an inverse Gamma prior, i.e., $\sigma_j^2 \sim$ Inv-Gamma$(a, b)$ with hyperparameters $a$ and $b$. Given $\sigma_{1:J}^2$, for conventional Bayesian regression mixture model, $\boldsymbol{\beta}_{1:J}$ can be drawn from Gaussian distribution $N(\mathbf{m}_\beta, \sigma_j^2 \mathbf{V}_\beta)$ with hyperparameters $\mathbf{m}_\beta$ and $\mathbf{V}_\beta$.

As is described above, using a mixture model to approximate a learning model, for any data sample we can identify a regression model to best approximate the prediction of that sample. This is due to the fact that a mixture model can be interpreted as arising from a clustering procedure which depends on the underlying latent component indicators $z_{1:n}$. For each observation $(\mathbf{x}_i, y_i)$, $z_i = j$ indicates that the observation was generated from the $j$-th Gaussian distribution, i.e., $y_i | z_i = j \sim N(\mathbf{x}_i \boldsymbol{\beta}_j, \sigma_j^2)$ with $P(z_i = j) = \pi_j$.

**Dirichlet Process Mixture Model with Multiple Elastic Nets.** Recall that a conventional mixture model has difficulty not only in dealing with high dimensional data and highly correlated features but also in handling different types of data heterogeneity. We modify the conventional mixture model by resetting the prior distribution of $\boldsymbol{\beta}_{1:J}$ to realize multiple elastic nets. Specifically, we first define mixture distribution

$$P(\boldsymbol{\beta}_j | \lambda_{1,1:K}, \lambda_{2,1:K}, \sigma_j^2) = \sum_{k=1}^K w_k f_k(\boldsymbol{\beta}_j | \lambda_{1,k}, \lambda_{2,k}, \sigma_j^2), \tag{4}$$

where $K$ denotes the total number of component distributions, and $w_{1:K}$ represent component probabilities with $\sum_{k=1}^K w_k = 1$. Let $w_k's$ follow a Dirichlet distribution, i.e., $w_1, w_2, \cdots, w_K \sim$ Dir$(1/K)$. Since we add elastic net regularization to the regression coefficient $\boldsymbol{\beta}_{1:J}$, instead of the aforementioned normal distribution, we adopt the Orthant Gaussian distribution as the prior distribution according to [9]. To be specific, each $\boldsymbol{\beta}_k$ follows a Orthant Gaussian prior, whose density function $f_k$ can be defined as

$$f_k\left(\boldsymbol{\beta}_j | \lambda_{1,k}, \lambda_{2,k}, \sigma_j^2\right) \propto \boldsymbol{\Phi}\left(\frac{-\lambda_{1,k}}{2\sigma\sqrt{\lambda_{2,k}}}\right)^{-p} \times \sum_{\mathbf{Z} \in \mathcal{Z}} N\left(\boldsymbol{\beta}_j \,\Big|\, -\frac{\lambda_{1,k}}{2\lambda_{2,k}}\mathbf{Z}, \frac{\sigma_j^2}{\lambda_{2,k}}\mathbf{I}_p\right) \mathbf{1}(\boldsymbol{\beta}_j \in \mathcal{O}_{\mathbf{Z}}). \tag{5}$$

Here, $\lambda_{i,k}$ $(i = 1, 2)$ is a pair of parameters which controls lasso and ridge regression for the $k$-th component, respectively. We set both to follow Gamma conjugate prior with $\lambda_{1,k} \sim$ Gamma$(R, V/2)$ and $\lambda_{2,k} \sim$ Gamma$(L, V/2)$, where $R$, $L$, and $V$ are hyperparameters. $\boldsymbol{\Phi}(\cdot)$ is the cumulative distribution function of the univariate standard Gaussian distribution, and $\mathcal{Z} = \{-1, +1\}^p$ is a collection of all possible $p$-vectors with elements $\pm 1$. Let $Z_l = 1$ for $\beta_{jl} \geq 0$ and $Z_l = -1$ for $\beta_{jl} < 0$. Then, $\mathcal{O}_{\mathbf{Z}} \subset \mathbb{R}^p$ can be determined by vector $\mathbf{Z} \in \mathcal{Z}$, indicating the corresponding orthant.

Given the prior distribution of $f_k$ defined in (5), it is difficult to compute the posterior distribution and sample from it. To obtain a simpler form, we use the mixture representation of the prior distribution (5). To be specific, we introduce a latent variable $\boldsymbol{\tau}_{1:p}$ and rewrite the (5) into the following hierarchical form[2]

$$\boldsymbol{\beta}_j \,|\, \boldsymbol{\tau}_j, \sigma_j^2, \lambda_{2,c_j} \sim N\left(\boldsymbol{\beta}_j \,\Big|\, 0, \frac{\sigma_j^2}{\lambda_{2,c_j}}\mathbf{S}_{\boldsymbol{\tau}_j}\right), \text{and} \tag{6}$$

$$\boldsymbol{\tau}_j \,|\, \sigma_j^2, \lambda_{1,c_j}, \lambda_{2,c_j} \sim \prod_{l=1}^p \text{Inv-Gamma}_{(0,1)}\left(\tau_{jl} \,\Big|\, \frac{1}{2}, \frac{1}{2}\left(\frac{\lambda_{1,c_j}}{2\sigma_j\sqrt{\lambda_{2,c_j}}}\right)^2\right), \tag{7}$$

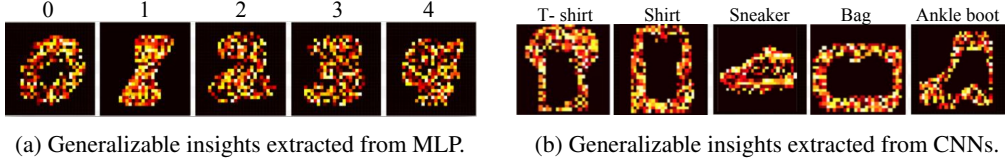

(a) Generalizable insights extracted from MLP.　　　(b) Generalizable insights extracted from CNNs.

Figure 1: The illustration of Generalizable insights extracted from the MLP trained for recognizing handwritten digits and the CNNs for fitting the Fashion-MNIST dataset. Each pattern contains 150 pixels, the importance of which is illustrated by the heat map. Due to the space limit, the results of other categories are shown in supplementary material.

where $\boldsymbol{\tau}_j \in \mathbb{R}^p$ denotes latent variables and $\mathbf{S}_{\boldsymbol{\tau}_j} \in \mathbb{R}^{p \times p}$, with $\mathbf{S}_{\boldsymbol{\tau}_j} = \text{diag}(1 - \tau_{jl})$ for $l = 1, \cdots, p$. Similar to component indicator $z_i$ introduced in the previous section, here, we introduce a set of latent regularization indicators $c_{1:J}$. For each parameter $\beta_j$, $c_j = k$ indicates that parameter follows distribution $f_k(\cdot)$ with $P(c_j = k) = w_k$.

**Posterior Computation and Post-MCMC Analysis.** We develop a customized MCMC method involving a combination of Gibbs sampling and Metropolis-Hastings algorithm for parameter inference [28]. Basically, it involves augmentation of the model parameter space by the aforementioned mixture component indicators $z_{1:n}$ and $c_{1:J}$. These indicators enable simulation of relevant conditional distributions for model parameters. As the MCMC proceeds, they can be estimated from relevant conditional posteriors and thus we can jointly obtain posterior simulations for model parameters and mixture component indicators. We provide the details of posterior distribution and the implementation of updating the parameters in the supplementary material. Considering that fitting a mixture model with MCMC suffers from the well-known label switching problem, we use an iterative relabeling algorithm introduced in [3].

# 4 Evaluation

Recall that the motivation of our proposed method is to increase the transparency for complex ML models so that users could leverage our approach to not only understand an individual decision (explainability) but also to obtain insights into the strength and vulnerabilities of the target model (scrutability). The experimental evaluation of the proposed method thus focuses on the aforementioned two aspects – scrutability and explainability.

## 4.1 Scrutability

**Methodology.** As a first step, we utilize Keras [2] to train an MLP on MNIST dataset [16] and CNNs to classify clothing images in Fashion-MNIST dataset [34] respectively. These machine learning methods represent the techniques most commonly used for the corresponding classification tasks. We trained these model to achieve more than decent classification performance. We then treat these two models as our target models and apply our proposed approach to establish scrutability.

We define the scrutability of an explanation method as the ability to distill generalizable insights from the model under examination. In this work, generalizable insights refer to feature importance inferences that could be generalized across all cases. Admittedly, the fidelity of our proposed solution to the target model is an important prerequisite to any generalizable insights our solution extracts. In this section, we carry out experiments to empirically evaluate the fidelity while also demonstrating scrutability of our solution. We apply the following procedures to obtain experimentation data.

1. Construct bootstrapped samples from the training data and nullify the top important pixels identified by our approach among positive cases while replacing the same pixels in negative cases with the mean value of those features among positive samples.

2. Apply random pixel nullification/replacement to the same bootstrapped samples used in previous step from the training data.

3. Construct test cases that register positive properties for the top important pixels while randomly assign value for the remaining pixels.

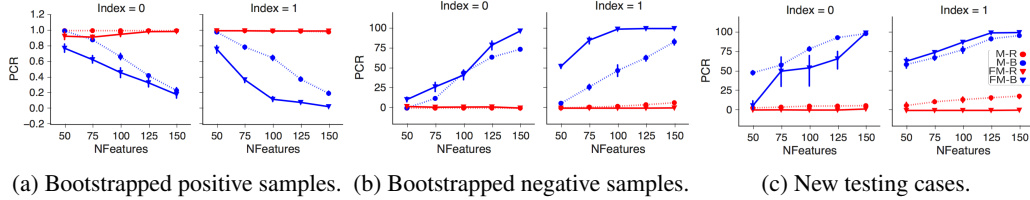

(a) Bootstrapped positive samples. (b) Bootstrapped negative samples. (c) New testing cases.

Figure 2: Results of fidelity validation. Note that `PCR` in y-axis denotes positive classification rate and `NFeature` in x-axis refers to number of features. In the legend, `B` indicates selecting features through our Bayesian approach and `R` represents selecting features through random pick. `M` and `FM` denote datasets MNIST and Fashion-MNIST respectively. Due to the space limit, the results of other categories are shown in supplementary material.

4. Construct randomly created test cases (*i.e.*, assigning random value to all pixels) as baseline samples for the new test cases.

We then compare the target model classification performance among synthetic samples crafted via procedures mentioned above. The intuition behind this exercise is that if the fidelity/scrutability of our proposed solution holds, we should be able to see significant impact on the classification accuracy. Moreover, the magnitude of the impact should significantly outweigh that observed from randomly manipulating features. In the following, we describe our experiment tactics and findings in greater details.

**Experimental Results.** Figure 1 illustrates the generalizable insights (*i.e.*, important pixels in MNIST and Fashion-MNIST datasets) that our proposed solution distilled from the target MLP and CNNs models, respectively. To validate the faithfulness of these insights and establish fidelity of our proposed solution, we conduct the following experiment.

First, bootstrapped samples, each contains a random draw of 30% of the original cases, are constructed from the MNIST and Fashion-MNIST datasets. For cases that are originally identified as positive for corresponding classes by the target models (*i.e.*, MLP and CNNs), we nullify top 50/75/100/125/150 important features identified by our proposed solution respectively, while forcing the value of corresponding features in the negative samples equal to the mean value of those among the positive samples. These manipulated cases are then supplied to the the target model and we measure the proportion of cases that those models would classify as positive under each condition. In addition, we apply the same perturbations on randomly selected 50/75/100/125/150 features in the same bootstrapped sample and measure the target model's positive classification rate after the manipulation as a baseline for comparison. We repeat such a process for 50 times for both datasets to account for the statistical uncertainty in the measured classification rate.

Figure 3a, Figure 3b and supplementary material showcase some of the aforementioned bootstrapped samples. Figure 2a and Figure 2b summarize the experimental results we obtain from the procedures mentioned above. As is illustrated in both figures, the classification rates of the target models on these perturbed samples are impacted dramatically once we start manipulating top 50/75 important features (*i.e.*, around 9% of the pixels in each image) identified by our proposed solution in these images. However, we do not observe any significant impact to the model's classification performance if we randomly perturb the same number of pixels. Non-overlapping 95% confidence intervals of the post-manipulation classification performance also reveal that the impact of these top features is significantly greater than the features selected at random. Moreover, the fact that we start observing dramatic impact in the target models' classification performance after we manipulate less than 9% of the total features justifies the faithfulness of our proposed approach to the ML models under examination.

To further validate the fidelity of the insights illustrated in Figure 1, we construct new testing cases based on top 50/75/100/125/150 pixels deemed important by our proposed solution respectively and measure the proportion of these testing samples that are classified as positive cases by the target models. We also create testing cases by randomly filling 50/75/100/125/150 pixels within the images and measure the positive classification rate as a baseline. The intuition behind this exercise is that, similar to the experiments described earlier, we would like to see significantly higher positive classification rates leveraging the insights from our proposed solution than creating cases around randomly selected pixels. In Figure 3c and supplementary material, we showcase some insights

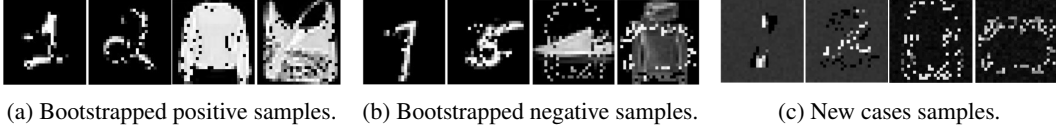

(a) Bootstrapped positive samples.  (b) Bootstrapped negative samples.  (c) New cases samples.

Figure 3: Samples manipulated or crafted for scrutability evaluation.

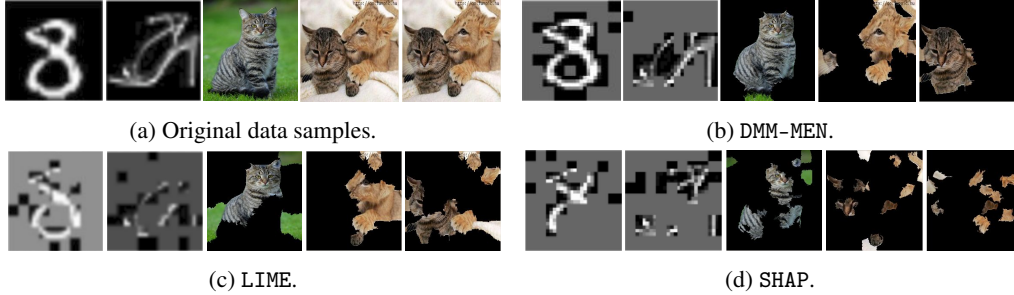

(a) Original data samples.                    (b) DMM-MEN.

(c) LIME.                                      (d) SHAP.

Figure 4: The examples explaining individual predictions obtained from MLP and CNNs. It should be noted that, since the images in MNIST and Fashion-MNIST has black background, to better illustrate the difference, we change segments of these images to grey if they are not selected.

driven testing cases. As is shown in Figure 2c, insights driven testing cases have much higher success rates than the cases created around random pixels. In fact, we observe that even if we randomly fill 150 pixels (which is close to 20% of the pixels in an image), the positive classification rate remains extremely low across classes. On the contrary, we notice that with the cases created based on the top 50 important pixels (*i.e.*, 9% of all pixels in an image) deemed by our solution, we could already achieve around 50% success rate. For some specific outcome categories, we could even achieve a much higher success rate.

It is worth noting that aforementioned experiments also unveil the vulnerabilities and sensitivities of the target MLP and CNNs models. It does not seem to matter if a handwritten digit or a fashion product is visually recognizable in an image, the model will classify it to the corresponding category with a high confidence as long as the important features indicated in the heat map are filled with greater values (see Figure 3b). In other words, both the MLP and CNNs models evaluated in this study are very sensitive to these pixels but could also be vulnerable to pathological image samples crafted based on such insights. Figure 3a and Figure 3c are two additional examples. A sample (Figure 3a) might carry the right semantics, the learning model still might be blind to that sample if the pixels corresponding to important features are filled with smaller values. On the other hand, a very noisy sample (Figure 3c) could still be correctly classified as long as the pixels corresponding to important features are assigned with decent values.

## 4.2 Explainability

Our proposed solution does not only extract generalizable insights from the target models but also demonstrate superior performance in explaining individual decisions. To illustrate its superiority, we compare our approach with a couple of state-of-the-art explainable approaches, namely LIME and SHAP. In particular, we evaluate the explainability of these approaches by comparing the explanation feature maps and more importantly quantitatively measuring their relative superiority in identifying influential features in individual decisions.

As is introduced in the aforementioned section, we also evaluate the explainability of our proposed solution on the VGG16 model [27] trained from ImageNet dataset [5]. Due to the ultra high dimensionality concern, which we will discuss in the following section, we adopt the methodology in [23] to generate data to explain individual decisions. More specifically, we create a new dataset by randomly sampling around the data sample that needed to be explained, reducing the dimensionality of the newly crafted dataset by certain dimension reduction method [23] and fitting the approximation model.

Table 1: Quantitative evaluation results of explainability

| | DMM-MEN | | LIME | | SHAP | |
|---|---|---|---|---|---|---|
| | Probability (Confidence Interval) | Accuracy | Probability (Confidence Interval) | Accuracy | Probability (Confidence Interval) | Accuracy |
| MNIST | 99.89% (99.74%, 100%) | 100% | 99.84% (99.69%, 100%) | 99.95% | 94.01% (93.99%, 94.03%) | 94.10% |
| Fashion-MNIST | 97.59% (97.32%, 97.89%) | 100% | 93.49% (92.92%, 94.07%) | 98.32% | 86.03% (85.23%, 86.65%) | 90.10% |
| ImageNet | 69.36% (47.88%, 90.18%) | 85.6% | 47.46% (31.34%, 68.58%) | 66.05% | 7.85% (5.88%, 28.82%) | 14.20% |

Figure 4a and supplementary material illustrate ten handwritten digits and ten fashion products randomly selected from each of the classes in MNIST and Fashion-MNIST datasets, respectively. We apply our solution as well as LIME and SHAP to each of the images shown in the figure and then select and highlight the top 20 segments that each approach deems important to the decision made by deep neural network classifiers. The results are presented in Figure 4b, Figure 4c, Figure 4d and supplementary material for our approach, LIME and SHAP, respectively. As we can observe in these figures, our approach nearly perfectly highlights the contour of each digit and fashion product, whereas LIME and SHAP identify only the partial contour of each digit and product and select more background parts than our approach.

Figure 4a also has two images we randomly selected from ImageNet dataset. The left image has only one object and the other image has two. Figure 4b to Figure 4d demonstrate the top 10 segments pinpointed by three explanation techniques. The results shown in these figures are consistent with those of MNIST and Fashion-MNIST. More specifically, the proposed approach can precisely highlight the object in the images, while the other approaches only partly identify the object and even select some background noise as important features. In order to evaluate the fidelity of these explanation results, we input these feature images back to VGG16 and record the prediction probabilities of the true labels (tiger cat, lion and tiger cat). Figure 4b achieved the highest probabilities on each feature map, which from the left to right are $93.20\%$, $78.51\%$ and $92.70\%$. Note that in the fourth image of Figure 4b, while identifying a lion in the image, our approach highlights the moustache of the cat, which seems like a wrong selection. However, if we exclude this part from the image, the probability of the object belonging to lion drops from $78.51\%$ to $20.31\%$. This result showcases a false positive of VGG16 and indicates that we can still find the weakness of the target model even from the individual explanations.

To further quantify the relative performance in explainability, we also conduct the following experiment. First we randomly select 10000 data samples from aforementioned datasets. Then, we apply our approach as well as two state-of-the-art solutions (*i.e.*, LIME and SHAP) to extract top 20 important segments (top 10 segments for ImageNet dataset). We then manipulate these samples based on the segments identified via three approaches. To be specific, we only keep the top important pixels intact while nullifying the remaining pixels and supply these manipulated samples to the target models and evaluate the classification accuracy. Table 1 shows the accuracy of these feature images being classified to the corresponding truth categories as well as the means and the $95\%$ confidence interval of the prediction probabilities. The results indicate that our approach offers better resolution and more granular explanations to individual predictions. One possible explanation is that both LIME and SHAP assume the local decision boundary of the target model to be linear while the proposed approach conducts the variable selection by applying a non-linear approximation.

It is known that Bayesian non-parametric models are computationally expensive. However, It does not mean that we cannot use the proposed approach in the real-world applications. In fact, we have recorded the latency of the proposed approach on explaining individual samples in three datasets. The running times are for MNIST, Fashion-MNIST and ImageNet are 37.5s, 44s and 139.2s, respectively. As to approximating the global decision boundary, the running times are 105 mins on MNIST and 115 mins on Fashion-MNIST. It is believed that the latency of our approach is still within the range of normal training time for complex ML models.

## 5 Discussion

**Scalability.** As is shown in Section 4, our proposed solution does not impose incremental challenge on scalability. We can still further accelerate the algorithm to improve its scalability. More specifically,

current advances in Bayesian computation approaches allow the MCMC methods to be used for big data analysis, such as adopting Bootstrap Metropolis–Hastings Algorithm [19], applying divide and conquer approaches [30] and even taking advantage of GPU programming to speed up the computation [31].

**Data Dimensionality.** Our evaluation described in Section 4 indicates that the proposed solution (`DMM-MEN`) could extract generalizable insights even from high dimensional data (*e.g.* Fashion MNIST). However, when it comes to ultra high-dimensional data, getting generalizable insights could still be a challenge. One obvious reason is that we do not have sufficient data to infer all the parameters. More importantly, even if we had enough data, it would be very computationally expensive. Arguably, one solution is to reduce the dimensionality of such ultra high dimensional data while preserving the original data distribution. However, take ImageNet dataset as an example. Even the state-of-the-art dimensionality reduction methods (*i.e.,* the one used in [23]) could not satisfactorily preserve the whole data distribution. This indeed speaks to the limitation of our proposed solution in extracting generalizable insights when it comes to specific datasets. Nevertheless, it does not affect our solution's ability in precisely explaining individual predictions even when it comes to ultra high dimensional data. As is shown in Section 4, our solution significantly outperforms the state-of-the-art solutions in explaining individual decisions made on ultra-high dimensional data samples.

**Other Applications and Learning Models.** While we evaluate and demonstrate the capability of our proposed technique only on the image recognition using deep learning models, the proposed approach is not limited to such a learning task and models. In fact, we also evaluated our technique on other learning tasks with various learning models. We observed the consistent superiority in extracting global insight and explaining individual decisions. Due to the space limit, we specify those experiment results in our supplementary material submitted along with this manuscript.

## 6 Conclusion and Future Work

This work introduces a new technical approach to derive generalizable insights for complicated ML models. Technically, it treats a target ML model as a black box and approximates its decision boundary through `DMM-MEN`. With this approach, model developers and users can approximate complex ML models with low errors and obtain better explanations of individual decisions. More importantly, they can extract generalizable insights learned by a target model and use it to scrutinize model strengths and weaknesses. While our proposed approach exhibits outstanding performance in explaining individual decisions, and provides a user with an ability to discover model weaknesses, its performance may not be good enough when applied to interpreting temporal learning models (*e.g.*, recurrent neural networks). This is due to the fact that, our approach takes features independently whereas time series analysis deals with features temporally dependent. As part of the future work, we will therefore equip our approach with the ability of dissecting temporal learning models.

**Acknowledgments** We gratefully acknowledge the funding from NSF grant CNS-1718459 and the support of NVIDIA Corporation with the donation of the GPU. We also would like to thank anonymous reviewers, Kaixuan Zhang, Xinran Li and Chenxin Ma for their helpful comments.

## Footnotes

[1] For multi-class classification tasks, this work approximates each class separately, and thus $\mathbf{X}$ denotes the samples in the same class and $g(\mathbf{X})$ represents the corresponding predictions. Given that $\mathbf{y}$ is a probability vector, we conduct logit transformation before fitting a regression mixture model.

[2]More details about the derivation of the scale mixture representation and the proof of equivalence can be found in [9, 18].

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
