[Supplementary Material]

# Supplementary Material: Explaining Deep Learning Models – A Bayesian Non-parametric Approach

## 1 Implementation

### 1.1 Posterior Distribution

We present the full posterior of the proposed approach as follows. The following parameters are updated via Gibbs Sampling. First,

$$P(z_i = j \mid \mathbf{x}_i, y_i, \boldsymbol{\beta}_j, \sigma_j^2) \propto P(y_i \mid \mathbf{x}_i, \boldsymbol{\beta}_j, \boldsymbol{\tau}_j, \sigma_j^2) P(z_i = j) \,, \tag{1}$$

and

$$u_j \Big| \mathbf{X}, \mathbf{y} \sim \text{Beta}(a_j + 1, \alpha + \sum_{l=j+1}^{J} a_l) \,, \tag{2}$$

where $a_j = \sum_{i=1}^{N} \mathbf{1}(z_i = j)$, for $j = 1, \cdots, J$ with $\mathbf{1}$ the indicator function. Then we let

$$\alpha \Big| u_{1:J-1} \sim \text{Gamma}(J + e + 1, f - \sum_{j=1}^{J-1} \log(1 - u_j)) \,, \tag{3}$$

$$P(c_j = k \mid \boldsymbol{\beta}_j, \sigma_j^2, \lambda_{2,k}) \propto P(\boldsymbol{\beta}_j \mid c_j = k, \sigma_j^2, \lambda_{2,k}) P(c_j = k) \,, \tag{4}$$

$$w_1, w_2, \cdots, w_K \Big| \frac{1}{K} \sim \text{Dir}\left( \frac{1}{K} + b_1 - 1, \frac{1}{K} + b_2 - 1, \cdots, \frac{1}{K} + b_K - 1 \right) \,, \tag{5}$$

where $b_k = \sum_{j=1}^{J} \mathbf{1}(c_j = k)$, for $k = 1, \cdots, K$, and

$$\boldsymbol{\beta}_j \mid \boldsymbol{\tau}_j, \lambda_{2,k}, \mathbf{X}, \mathbf{y} \sim N(\mathbf{R}_{\boldsymbol{\tau}_j} \mathbf{X}_{\mathcal{O}_j}^{\mathrm{T}} \mathbf{y}_{\mathcal{O}_j}, \sigma_j^2 \mathbf{R}_{\boldsymbol{\tau}_j}) \,, \tag{6}$$

where $\mathbf{R}_{\boldsymbol{\tau}_j} = (\mathbf{X}_{\mathcal{O}_j}^{\mathrm{T}} \mathbf{X}_{\mathcal{O}_j} + \lambda_{2,c_j} \mathbf{S}_{\boldsymbol{\tau}_j}^{-1})^{-1}$. We define $\mathcal{O}_j = \{i | z_i = j\}$. Then, $\mathbf{X}_{\mathcal{O}_j}$ represents the matrix composed by the vectors $\mathbf{x}_i$ whose subscript belongs to the set $\mathcal{O}_j$ and $\mathbf{y}_{\mathcal{O}_j}$ represents the vector, where each element is $y_i$ whose subscript also belongs to the set $\mathcal{O}_j$. By the change of variable, let $s_{jl} = \tau_{jl}/(1 - \tau_{jl})$ to ensure that $\tau_{jl}$ falls into the interval $(0, 1)$, then we have

$$s_{jl} \mid \boldsymbol{\beta}_j, \sigma_j^2, \lambda_{1,c_j}, \lambda_{2,c_j} \sim \text{Inv-Gamma}\left( \frac{\lambda_{1,c_j}}{2\lambda_{2,c_j} |\boldsymbol{\beta}_j|}, \frac{\lambda_{1,c_j}^2}{4\lambda_{2,c_j}} \sigma_j^2 \right) \,. \tag{7}$$

The other three sets of variables $\sigma_{1:J}^2$, $\lambda_{1,1:K}$, $\lambda_{2,1:K}$ are updated by Metropolis-Hasting Sampling. The posterior and proposal distribution of variables in each set are as follows. First,

$$P(\sigma_j^2 \mid \boldsymbol{\beta}_j, \boldsymbol{\tau}_j, \mathbf{X}, \mathbf{y}) \propto \prod_{i=1}^{a_j} P(y_i \mid \mathbf{x}_i \boldsymbol{\beta}_j, \sigma_j^2) P(\boldsymbol{\beta}_j \mid \sigma_j^2, \boldsymbol{\tau}_j, \lambda_{2,c_j}) P(\boldsymbol{\tau}_j \mid \sigma_j^2, \lambda_{1,c_j}, \lambda_{2,c_j}) P(\sigma_j^2) \,. \tag{8}$$

Then the proposed distribution follows

$$\prod_{i=1}^{a_j} P(y_i \mid \mathbf{x}_i \boldsymbol{\beta}_j, \sigma_j^2) P(\boldsymbol{\beta}_j \mid \sigma_j^2, \boldsymbol{\tau}_j, \lambda_{2,c_j}) P(\sigma_j^2) \propto \text{Inv-Gamma}(a_N, b_N) \,, \tag{9}$$

where $a_N = a + (a_j + 1)/2$ and $b_N = b + (\sum_{i=1}^{a_j} (g_i - \mathbf{x}_i \boldsymbol{\beta}_j)^2 + \boldsymbol{\beta}_j^{\mathrm{T}} \mathbf{S}_{\boldsymbol{\tau}_j}^{-1} \boldsymbol{\beta}_j \lambda_{2,c_j})/2$.

For $\lambda_{1,k}$ and $\lambda_{2,k}$,

$$P(\lambda_{1,k}, \lambda_{2,k} \mid \boldsymbol{\beta}_j, \boldsymbol{\tau}_j) \propto \prod_{j=1}^{b_k} P(\boldsymbol{\beta}_j \mid \boldsymbol{\tau}_j, \lambda_{2,k}, \sigma_j^2) P(\boldsymbol{\tau}_j \mid \lambda_{1,k}, \lambda_{2,k}) P(\lambda_{1,k}, \lambda_{2,k}) \,. \tag{10}$$

We use a truncated normal distribution to represent the proposed distribution, that is,

$$\lambda_{1,k} \sim \text{TN}(\lambda_{1,k}, v_{1,N}), \quad \lambda_{2,k} \sim \text{TN}(\lambda_{2,k}, v_{2,N}) \,, \tag{11}$$

where TN stands for truncated normal distribution, and $v_{1,N}$ and $v_{2,N}$ are the hyperparameters.

## 1.2 Training Algorithm

The algorithm of estimate the parameters of the proposed model is shown in Table 1. We implement the algorithm using R and the code is available at https://github.com/Henrygwb/dmm-men.

---

**Algorithm:** Customized MCMC for Parameter Inference

**Input:** Data Matrix $\mathbf{X}$, corresponding prediction $\mathbf{y}$, iteration number $TT$, mixture component number $J$, elastic nets number $K$, hyper-parameters $e$, $f$, $a$, $b$, $L$, $R$, $V$.

**Initialization:** Sampling $\alpha^{(1)}$, $\mu_{1:J-1}^{(1)}$, $\pi_{1:J-1}^{(1)}$, $z_{1:n}^{(1)}$, $\lambda_{1,1:K}^{(1)}$, $\lambda_{2,1:K}^{(1)}$, $w_{1:K}^{(1)}$, $c_{1:J}^{(1)}$, $\sigma_{1:J}^{2}{}^{(1)}$, $\boldsymbol{\beta}_{1:J}^{(1)}$, $\boldsymbol{\tau}_{1:J}^{(1)}$ from the prior distributions introduced in the paper.

**for** $tt = 2, ..., TT$ **do**

    Updating the following parameters with Gibbs sampling:

    $z_{1:n}^{(tt)}$,

    $\mu_{1:J-1}^{(tt)}$,

    $\pi_{1:J-1}^{(tt)}$,

    $\alpha^{(tt)}$,

    $c_{1:J}^{(tt)}$,

    $w_{1:K}^{(tt)}$,

    $\boldsymbol{\beta}_{1:J}^{(tt)}$,

    $\boldsymbol{s}_{1:J}^{(tt)}$ and $\boldsymbol{\tau}_{1:J}^{(tt)}$.

    Updating the following parameters by Metropolis-Hasting sampling:

    $\sigma_{1:J}^{2}{}^{(tt)}$,

    $\lambda_{1,1:K}^{(tt)}$ and $\lambda_{2,1:K}^{(tt)}$.

**end for**

**Output:** $\alpha^{(TT)}$, $\mu_{1:J-1}^{(TT)}$, $\pi_{1:J-1}^{(TT)}$, $z_{1:n}^{(TT)}$, $\lambda_{1,1:K}^{(TT)}$, $\lambda_{2,1:K}^{(TT)}$, $w_{1:K}^{(TT)}$, $c_{1:J}^{(TT)}$, $\sigma_{1:J}^{2}{}^{(TT)}$, $\boldsymbol{\beta}_{1:J}^{(TT)}$, $\boldsymbol{\tau}_{1:J}^{(TT)}$.

---

Table 1: The proposed customized MCMC algorithm to train the DMM-MEN model. Note that the details of posterior distributions for Gibbs sampling and the proposal distributions for Metropolis-Hasting sampling can be found in the Section 1.1.

| | Neural Network Structure | Activation | Optimizer | Learning Rate | Regularization | Batch | Epoch |
|---|---|---|---|---|---|---|---|
| DNN | 784-512-512-10 | Relu | RMSprop | 0.001 | Dropout (0.2) | 128 | 50 |
| CNN | Shown in Table 3 | Relu | RMSprop | 0.001 | × | 500 | 30 |

Table 2: Hyperparameters of DNN trained from MNIST and CNN trained from Fashion-MNIST.

| Layer Type | CNN Architecture |
|---|---|
| Convolotional | 32 filters ($5 \times 5$) |
| Max Pooling | $2 \times 2$ |
| Convolotional | 64 filters ($5 \times 5$) |
| Max Pooling | $2 \times 2$ |
| Softmax | 10 |

Table 3: Architecture of CNN model. Note that we build the model according to one of baselines shown in [3].

# 2 Hyperparameters

## 2.1 Hyperparameters of target models

The hyperparameters of the MLP trained on MNIST dataset and the CNNs trained to classify fashion products in Fashion-MNIST dataset are shown in Table 2. Note that we trained the target deep learning models to achieve the state-of-the-art classification performance on the original datasets, the test accuracy of which are $98.32\%$ on MNIST and $91.24\%$ on Fashion-MNIST.

## 2.2 Hyperparameters of proposed technology

The hyperparameters of the proposed `DMM-MEN` on each target machine learning model are shown in Table 4.

| Datasets | Target models | J | e | f | K | R | L | V | a | b | $v_{1_N}$ | $v_{2_N}$ |
|---|---|---|---|---|---|---|---|---|---|---|---|---|
| 'comp.sys' | Random Forest | 5 | 1 | 1 | 3 | 4 | 1 | 1 | 0.5 | 0.5 | 2 | 2 |
| | SVM | 5 | 5 | 1 | 3 | 5 | 1 | 1 | 1 | 1 | 2 | 2 |
| MNIST | MLP | 6 | 5 | 1 | 3 | 2.5 | 1 | 1 | 0.5 | 0.5 | 2 | 2 |
| Fashion-MNIST | CNNs | 10 | 5 | 1 | 3 | 2.5 | 1 | 1 | 0.5 | 0.5 | 2 | 2 |
| ImageNet | CNNs | 10 | 5 | 1 | 3 | 2.5 | 1 | 1 | 0.5 | 0.5 | 2 | 2 |

Table 4: Hyperparameters of proposed methods for all the target models

In Table 4, $J$ is the upper bound of the mixture components, $e$ and $f$ are the hyperparameters of Gamma($e$, $f$) and $a$, $b$ are the hyperparameters of Inv-Gamma($a$, $b$). $K$ is the total number of elastic-net. $R$, $L$, $V$ are the hyperparameters of Gamma($R$, $V/2$) and Gamma($L$, $V/2$). $v_{1_N}$ and $v_{2_N}$ are the hyperparameters of the truncated normal distribution in (11).

## 2.3 Experimental Results on Image Recognition.

## 2.4 Scrutability

Figure 1 demonstrates the generalizable insights for all of the categories in MNIST and Fashion-MNIST. Similar to **Section 4**, we highlights the most important 150 pixels in each category. Figure 7 shows the fidelity test results of all of the categories.

Figure 8 - Figure 10 showcase more examples of fidelity testing samples generated from MNIST dataset. Samples generated from Fashion-MNIST are shown in Figure 11 - Figure 13. Actually, we can automatically generate a large bunch of fidelity testing samples according to the method introduced in the **Section 4**. These samples can not only be used to evaluate the fidelity of the proposed approach, but also serve as adversarial samples (*i.e.*, Bootstrapped positive samples shown in 8 and 11) and pathology samples (*i.e.*, Bootstrapped negative samples shown in 9 and 12 and new

(a) Generalizable insights extracted from MLP.

(b) Generalizable insights extracted from CNNs.

Figure 1: The illustration of Generalizable insights extracted from the MLP trained for recognizing handwritten digits and the CNNs for fitting the Fashion-MNIST dataset.

(a) Handwritten digits randomly selected from MNIST and Fashion-MNIST datasets.

(b) Most influential pixels highlighted by DMM-MEN.

(c) Most influential pixels highlighted by LIME.

(d) Most influential pixels highlighted by SHAP.

Figure 2: The examples explaining individual predictions obtained from MLP and CNN. It should note that, to better illustrate the difference, we change pixels in gray if they are not selected.

testing samples shown in 10 and 13). To be specific, adversarial samples refer to samples that carry the right semantics but be classified to the wrong class by the target learning models. Pathology samples are samples that being correctly classified by the target learning models but do not contains the correct objects. Note that both of these samples explore the weakness of the target models; meanwhile the adversarial samples can be adopted to retrain the target model and improve its robustness.

## 2.5 Explainability

Figure 2 demonstrates more explanation results of MNIST and Fashion-MNIST datasets.

Note that in our evaluation, the LIME established better explanability than SHAP, which does not align with the theoretical results in [2]. The reason is that SHAP guarantees the best results by exploring nearly all of the possible feature combinations in the feature space. However, in practice, this is extremely hard to accomplish. In their implementation, SHAP conduct a limit number of combinations searching. It is highly likely that within this number of searching, SHAP is still not able to identify optimal combination of important features.

It is known that Bayesian non-parametric models are computationally expensive. However, It does not mean that we cannot use the proposed approach in the real-world applications. In fact, we have recorded the latency of the proposed approach on explaining individual samples in three datasets. The running times are for MNIST, Fashion-MNIST and ImageNet are 37.5s, 44s and 139.2s, respectively. As to approximating the global decision boundary, the running times are 105 mins on MNIST and 115 mins on Fashion-MNIST. It is believed that the latency of our approach is still within the range of normal training time for complex ML models.

| Random Forest | | SVM | |
|---|---|---|---|
| 'ibm.pc.hardware' | 'mac.hardware' | 'ibm.pc.hardware' | 'mac.hardware' |
| 'dos' | 'mac' | 'ide' | 'mac' |
| 'controller' | 'apple' | 'gateway' | 'Macintosh' |
| 'pc' | 'quadra' | 'dos' | 'quadra' |
| 'windows' | 'edu' | 'pc' | 'apple' |

Table 5: The keywords that our approach extracts, indicating the features most influential upon classifications.

Figure 3: An example text snippet categorized into 'ibm.pc.hardware' by random forest (RF) and SVM (SVM). The percentages shown in the table indicate the confidence of being categorized in 'ibm.pc.hardware'.

Figure 4: An example text snippet categorized into 'mac.hardware' by random forest (RF) and SVM (SVM). The percentages shown in the table indicate the confidence of being categorized in 'mac.hardware'.

# 3   Experimental Results on Text Mining.

Besides the experiments on MLP and CNNs shown in the **Section 4** of our paper, we also applied our method to machine learning models that are self explainable (*i.e.*, random forest and support vector machines) on text mining. Similar with **Section 4**, we first introduce the dataset used to train the random forest and support vector machines (SVM) models. Then, we demonstrate the scrutability and explainablity of our proposed method. Note that since these target models are self explainable, we donot need to test the fidelity of our proposed method by experiments. Actually, the fidelity of our model can be evaluated by simply comparing the important features extracted by our method with those identified from original models.

The dataset we use is a subset of news20 newsgroups dataset [1]:

**'comp.sys' newsgroups dataset** [1]: It is a collection of newsgroups posts containing 1,945 samples across 2 topics. The newsgroups posts are split into training and testing datasets based on the dates they have been posted.

## 3.1 Scrutability

Since SVM and random forest models are explainable by nature, we leverage this case to quantify our solution's faithfulness to a target model. To be specific, we identify top four influential words for each of the classification task using our solution, which are shown in Table 5, and compare them with the top four most weighted features in the original model. While the order of top four influential words vary slightly in our comparison against SVM, the words that our solution identifies match perfectly with those revealed in SVM and random forest model.

Figure 3 shows one classification example, in which both learning models classify the snippet into 'ibm.pc.hardware' with high confidence (94.20% and 93.47% for random forest and SVM, respectively). We replace word '*dos*' – important for both classifiers – with the words shown in Figure 3, and test each of the newly crafted snippets against both classifiers. The value shown in Figure 3 indicates the confidences of categorizing new snippets into 'ibm.pc.hardware'. We notice that by replacing '*dos*' with words that our solution deems important for another class (*i.e.*, 'mac.hardware'), we dramatically reduce the ML model's confidence in classifying the snippet under investigation as 'ibm.pc.hardware'. This again verifies the patterns that our approach extracts accurately reflect what are learned by both ML classifiers.

We also take an text snippet belonging to 'mac.hardware' – shown in Figure 4. In this example, both learning models classify the snippet into 'mac.hardware' with high confidence (99.40% and 99.90% for random forest and SVM, respectively). The words shown in Figure 4 are used to replace word '*Mac*'. We also test each of the newly crafted snippets against both classifiers. The value shown in Figure 4 indicates the confidences of categorizing new snippets into 'ibm.pc.hardware'. Similar to the results of 'mac.hardware', by replacing '*Mac*' with words that our solution deems important for 'mac.hardware', the confidences of being classified to 'ibm.pc.hardware' by ML model's are dramatically reduced. This again also verifies the important words that our approach extracts accurately reflect what are learned by both ML classifiers.

## 3.2 Explainablity

| | Random Forest | SVM | From: noah@apple.com (Noah Price) |
|---|---|---|---|
| DMM-MEN | **mac**<br>**apple**<br>**quadra**<br>**edu** | **mac**<br>**Macintosh**<br>**quadra**<br>**apple** | Subject: Re: How long do RAM SIMM's last?<br>Distribution: usa<br>Organization: (not the opinions of) Apple Computer, Inc<br>Lines: 12<br>In article <1993Apr11.234818.1755@ultb.isc.rit.edu>,<br>jek5036@ultb.isc.rit.edu (J.E. King) wrote: |
| LIME | **Macintosh**<br>**edu**<br>**apple**<br>**Re** | **apple**<br>**SIMM**<br>**Macintosh**<br>**about** | Doesn't a 1 MB SIMM have about 1024 * 1024 * 8 moving flip-flops?<br>…..<br>…..<br>…..<br>….. |
| SHAP | **mac**<br>**Macintosh**<br>**edu**<br>**apple** | **mac**<br>**apple**<br>**Macintosh**<br>**noah** | noah<br>〜〜〜〜〜〜〜〜〜〜〜〜〜〜〜〜〜〜〜〜〜〜〜<br>noah@apple.com　　　　Macintosh Hardware Design<br>...!{sun,decwrl}!apple!noah   (not the opinions of) Apple Computer, Inc. |

Figure 5: The examples explaining individual predictions obtained from random forest and SVM trained for classifying 'mac.hardware' news posts. Note that the text in bold indicate top-4 keywords most influential upon text classification.

Similar to what we observe in image recognition cases, our approach also outperforms LIME and SHAP in the context of text classification. Figure 5 illustrates one such example: the words highlighted are the most influential indicators for determining if the text snippet belongs to the category of 'mac hardware'. By applying both our approach, LIME and SHAP to the random forest and SVM classifiers, we can observe that the keywords highlighted by our approach is intuitively more distinguishable than those identified by LIME and SHAP.

Figure 6 demonstrates another examples. The words highlighted are the most important words for 'ibm.pc.hardware'. Results also indicate that proposed technology provides more distinguishable key words than those identified by LIME and SHAP. For example, 'SCSI' is a kind of computer interface, which also has been used to Macintosh. Therefore, it can not be treated as an indicator of 'ibm.pc.hardware'.

| | Random Forest | SVM | |
|---|---|---|---|
| DMM-MEN | **dos**<br>**controller**<br>**pc**<br>**windows** | **ide**<br>**gateway**<br>**dos**<br>**pc** | From: guyd@austin.ibm.com (Guy Dawson)<br>Subject: Re: IDE vs SCSI<br>Originator: guyd@pal500.austin.ibm.com<br>rganization: IBM Austin<br>Lines: 35<br>In article <1qlbrlINN7rk@dns1.NMSU.Edu>, bgrubb@dante.nmsu.edu (GRUBB)<br>writes: In PC Magazine |
| LIME | **ide**<br>**dos**<br>**controller**<br>**pc** | **ide**<br>**SCSI**<br>**pc**<br>**windows** | April 27, 1993:29 "Although SCSI is twice as fasst as ESDI, 20% faster than IDE, and support up to 7 devices its acceptance …<br>....<br>....<br>.... |
| SHAP | **dos**<br>**controller**<br>**ide**<br>**pc** | **ide**<br>**dos**<br>**pc**<br>**windows** | ....<br>I beleive this last bit is just plain wrong! SCSI-1 intergration is sited as another part of the MicroSoft Plug and play program.<br>Guy Dawson - Hoskyns Group Plc. |

Figure 6: The examples explaining individual predictions obtained from random forest and SVM trained for classifying 'ibm.pc.hardware' news posts.

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

(a) Bootstrapped positive samples.

(b) Bootstrapped negative samples.

(c) New testing cases.

Figure 7: Results of fidelity validation on each category of MNIST and Fashion-MNIST. `PCR` in y-axis denotes positive classification rate and `NFeature` in x-axis refers to number of features. In the legend, `B` indicates selecting features through our Bayesian approach and `R` represents selecting features through random pick. `M` and `FM` denote datasets MNIST and Fashion-MNIST respectively.

Figure 8: Bootstrapped positive samples of MNIST.

Figure 9: Bootstrapped negative samples of MNIST.

Figure 10: New testing cases of MNIST.

Figure 11: Bootstrapped positive samples of Fashion-MNIST.

Figure 12: Bootstrapped negative samples of Fashion-MNIST.

Figure 13: New testing cases of of Fashion-MNIST.