[Reviews · NeurIPS 2018]

Reviewer 1



I think the rebuttal is prepared very well. Although the assumption of a single component approximating the local decision boundary is quite strong, the paper nonetheless offers a good, systematic approach to interpreting black box ML systems. It is an important topic and I don't see a lot of studies in this area. Overview In an effort to improve scrutability (ability to extract generalizable insight) and explainability of a black box target learning algorithm the current paper proposes to use infinite Dirichlet mixture models with multiple elastic nets (DMM-MEN) to map the inputs to the predicted outputs. Any target model can be approximated by a non-parametric Bayesian regression mixture model. However, when samples exhibit diverse feature correlations, sparsity, and heterogeneity standard non-parametric Bayesian regression models may not be sufficients. The authors integrate multiple elastic nets to deal with diverse data characteristics. Extensive experiments are performed on two benchmark data sets to evaluate the proposed method in terms of scrutability and explainability. For scrutability a bootstrap sample of the training set is used to replace the most important pixels (as determined by the proposed algorithm) in images with random values. The intuition behind this exercise is that if the selected pixels are indeed strong features then the classification accuracy should suffer significantly and the degree of impact should outweigh other scenarios with randomly manipulated features. This intuition is demonstrated to be correct on both data sets. For explainability the proposed method is compared against two other methods from the literature (LIME and SHAP). The proposed method can identify parts of images that contribute to decisions with greater level accuracy than the other two techniques. Results suggest that the proposed technique can also be used to identify anomalies (inconsistent predictions of the target system) with some level of success. Technical Quality I would consider the technical quality to be good with some reservations. The main idea relies on the assumption that one component in the mixture model can be sufficient to explain a single prediction. What happens if a single component cannot approximate the local decision boundary near a single instance with an acceptable accuracy? In reality one would expect that many Normal components might be needed to explain a single prediction. So using a single component can make the prediction uninterpretable. What aspect of the model guarantee or stipulate sparse mixture components? The motivation behind using Orthant Gaussian prior on regression coefficients is not well justified. Does this really serve for its purpose (data dimensionality and heteregoneity)? What would happen if standard Gaussian prior was used as a prior? I also do not follow the need for truncation when the finite number of components can be easily determined during inference without any truncation especially when MCMC inference is used. Clarity The paper reads well with some minor issues outlined below (list is not exhaustive). Line 48 Most of the works that related to, "that" here is redundant Line 81 any learning models, "model" Line 320 when it comes specific datasets, "to" missing Line 320 it dose not affect, "does" Originality I would consider both the method (Dirichlet mixture models with multiple elastic nets) and the approach (evaluation in terms of scrutinability and explainability) quite original. Significance: I expect the significance of this work to be high as there is a dire need in the ML literature for models that can make outputs of complex learning systems more interpretable. Toward achieving this end the current paper proposes a systematic approach to explain and scrutinize the outputs of deep learning classifiers with promising results. Other Comments: Please define PCR (Principal Component Regression) and explain why PCR is used as opposed to classification accuracy

Reviewer 2



The goal of the paper is to provide a methodology to explain deep learning models in generality. The approach is based on Bayesian nonparametric (BNP) methods, more specifically on a BNP regression mixture model with (truncated) Dirichlet process mixing measure. The results of the linear regression model are used to interpret decide which features are important, and grouping them as explaining the decisions taken by the deep model. I have some concern with the prior distribution on the regression coefficients \beta_j's. This prior is defined in two different places, ie (4-5) and (6-7), but they are not consistent. could you comment why the variance for the prior on \beta_j is the same as the variance for the model, \sigma_j^2? commenting the choice for the covariance matrix in (6-7) would be useful, as it does not seem common practice. The authors should comment on the implementation of the MCMC algorithm, such as running time, mixing properties, convergence: insuring a good mixing of the chains in dimension p=28x28 does not seem trivial at all. Reproducibility of the experiments: a lot of (hyper)parameters are not specified in the model (including R, L, V in line 144) are not specified. Section 3 describes the technical details about the BNP mixture model. However, the link with the next sections is not clearly stated, so that it is difficult to see how scrutability and explainability are derived from the BNP model results. Additionally, some phrasings are vague and difficult to understand. See eg the paragraph on data dimensionality, lines 311-324.

Reviewer 3



The paper proposes approximating a complex ML model with a less complicated one for easier interpretation and proposes a specific model, a Bayesian non-parametric regression mixture model, to do this. Recent related papers that also propose ways of interpreting complex ML algorithms are • Piotr Dabkowski and Yarin Gal. Real time image saliency for black box classifiers. In Advances in Neural Information Processing Systems, pages 6970–6979, 2017. • Springenberg, J., et al. "Striving for Simplicity: The All Convolutional Net." ICLR (workshop track). 2015. • Koh, Pang Wei, and Percy Liang. "Understanding Black-box Predictions via Influence Functions." International Conference on Machine Learning. 2017. Especially the idea of approximating a complex model with a less complex one is not new. • Wu, Mike, et al. "Beyond sparsity: Tree regularization of deep models for interpretability." AAAI 2018 • Frosst, Nicholas, and Geoffrey Hinton. "Distilling a Neural Network Into a Soft Decision Tree." (2018). To my knowledge, the key innovation of this paper is using a Bayesian non-parametric regression mixture model, which allows for better performance of the interpretable model, one of the main weaknesses of less complex approaches. The experiments used to evaluate the method were extensive and compared with current popular approaches. Regarding clarity of the paper, there are some improvements still possible. This is especially visible in the explanation of the proposed approach. The specific restrictions and modifications to the model seem very ad-hoc and could be motivated better. 3.3 Technical details should bbe written clearer and more concise . F.e. “where alpha can be drawn from Gamma(e,f), a Gamma conjugate prior with hyperparameters e and f.” can be shortened to “where alpha can be drawn from Gamma(e,f)” Overall I think the paper proposes and evaluates a novel idea in a satisfactory way. Minor comments: Better labeling for figure 1 Edit after author rebuttal: I have updated my score to a strong 7. Ultimately I think the idea in the paper is novel and has potential. However, the presentation of this idea is unclear and vague enough to prevent an 8 for me.